# Evaluation of a Novel Synthetic Peptide Derived from Cytolytic Mycotoxin Candidalysin

**DOI:** 10.3390/toxins14100696

**Published:** 2022-10-11

**Authors:** Pedro Henrique de Oliveira Cardoso, Ana Paula de Araújo Boleti, Patrícia Souza e Silva, Lincoln Takashi Hota Mukoyama, Alexya Sandim Guindo, Luiz Filipe Ramalho Nunes de Moraes, Caio Fernando Ramalho de Oliveira, Maria Ligia Rodrigues Macedo, Cristiano Marcelo Espínola Carvalho, Alinne Pereira de Castro, Ludovico Migliolo

**Affiliations:** 1S-Inova Biotech, Programa de Pós-graduação em Biotecnologia, Universidade Católica Dom Bosco, Campo Grande 79117-900, Mato Grosso do Sul, Brazil; 2Laboratório de Purificação de Proteínas e suas Funções Biológicas, Unidade de Tecnologia de Alimentos e da Saúde Pública, Universidade Federal de Mato Grosso do Sul, Campo Grande 79070-900, Mato Grosso do Sul, Brazil

**Keywords:** *Candida albicans*, drug design, peptide toxin, neuroinflammation, multiactivity peptide

## Abstract

The importance of neuroinflammation in neurology is becoming increasingly apparent. In addition to neuroinflammatory diseases such as multiple sclerosis, the role of neuroinflammation has been identified in many non-inflammatory neurological disorders such as stroke, epilepsy, and cancer. The immune response within the brain involves the presence of CNS resident cells; mainly glial cells, such as microglia, the CNS resident macrophages. We evaluated the peptide Ca-MAP1 bioinspired on the *C. albicans* immature cytolytic toxin candidalysin to develop a less hemolytic peptide with anti-neuroinflammatory, antibacterial, and cytotoxic activity against tumor cells. *In silico* and *in vitro* studies were performed at various concentrations. Ca-MAP1 exhibits low hemolytic activity at lower concentrations and was not cytotoxic to MRC-5 and BV-2 cells. Ca-MAP1 showed activity against *Acinetobacter baumannii*, *Escherichia coli* ATCC, *E. coli* KPC, *Klebsiella pneumoniae* ATCC, *Pseudomonas aeruginosa*, and *Staphylococcus aureus* ATCC. Furthermore, Ca-MAP1 exhibits anti-neuroinflammatory activity in the BV-2 microglia model, with 93.78% inhibition of nitrate production at 18.1 µM. Ca-MAP1 presents cytotoxic activity against tumor cell line NCI-H292 at 36.3 μM, with an IC_50_ of 38.4 µM. Ca-MAP1 demonstrates results that qualify it to be evaluated in the next steps to promote the control of infections and provide an alternative antitumor therapy.

## 1. Introduction

Neuroinflammation is defined as the response of central nervous system (CNS) cells to infections, as well as to the infiltration of the brain and spinal cord by cells of the innate and adaptive immune system [1]. A prolonged or very intense inflammatory response in the CNS causes neuronal death by increasing levels of pro-inflammatory cytokines, proteases, glutamate, free radicals, and excessive activation of glial cells [1,2].

The origin of neuroinflammation can be multifactorial and includes infections, accumulation of toxic metabolites in obese or aging individuals, chemotherapy-related cognitive impairment, and peptides or proteins, as seen in various neurodegenerative diseases, Alzheimer’s disease, Parkinson’s disease, amyotrophic lateral sclerosis, ischemic injury, and multiple sclerosis [3,4].

The membrane component of Gram-negative bacteria, such as lipopolysaccharide (LPS), is a trigger of the inflammatory process that can cause an exacerbated reaction and induce the production of reactive molecules that directly damage cells by activating cascades of reactions that might cause sepsis and death; it is also related to neuroinflammation that can lead to the onset of neurodegenerative diseases. Forms of treatment that deal with both bacteria and LPS are highly desirable from a therapeutic viewpoint [5,6,7].

The search for a new form of treatment has led to the search for molecules called antimicrobial peptides (AMPs), especially those with a cationic helical structure, that act as mediators in cases of routine infections, hospital sepsis, or bacterial resistance by inhibiting the growth or destroying the membranes of invading cells [8,9]. Fungi are microorganisms that share many ecological niches with bacteria and end up competing with each other; this competition causes them to develop molecules, such as AMP, that can eliminate their competitors [10,11,12]. New molecules from organisms such as fungi are an alternative in the search for the control and/or combat of pathogenic organisms.

The pathogenic fungi *Candida albicans*, which grows in the mouth, digestive tract, and in the external portion of the reproductive system of most of the world population, can compete with human pathogenic bacteria and is capable of synthesizing a cationic cytolytic peptide toxin called candidalysin. Candidalysin in lower concentrations (1.5–15 µM) has already induced immunomodulatory effects such as DNA binding of c-Fos, G-CSF, and GM-CSF. Additionally, concentrations of 70 µM are already able to induce cell damage and damage-associated cytokines such as IL-1 α and IL-6 [13,14,15].

The cytolytic peptide toxin candidalysin is critical for the pathogenesis of the *C. albicans* fungi, thus showing the potential to be used in drug design to modulate its activity, because fungi can eliminate or inhibit other microorganisms when changing to its filamentous form and causing candidiasis [13,14,15]. The present study proposes to evaluate the anti-neuroinflammatory activity of synthetic peptides based on the protein sequence originating from the candidalysin of the fungus *C. albicans* as a new alternative molecule with potential against neurodegenerative diseases and bacterial infections, and also against cancer.

## 2. Results

### 2.1. Rational Design of the Ca-MAP1 Peptide

The generation of the derivate peptide was designed based on the parental peptide immature candidalysin (_NH2_-SIIGIIMGIIMGILGNIPQVIQIIMSIVKAFKGNKR-_COOH_), an important cytolytic toxin from *Candida albicans* [15], so that it should present at least 70% similarity to the parental peptide, preserving its physicochemical properties.

To perform the rational design of the derivates, a segment with half the number of amino acid residues of the parental peptide candidalysin primary sequence was chosen (Figure 1); the criteria to choose the segment were the presence of characteristics close to the antimicrobial peptides described in the literature, such as the charge from +2 to +4, percentage of hydrophobic residues close to 45 to 55%, and amphipathicity, thus being subjected to only four amino acid modifications to preserve its identity [16,17].

The derivate was defined as Ca-MAP1 (*Candida albicans*-Multiactivity Peptide 1), generated from the final segment of the candidalysin _17_IQIIMSIVKAFKGNKR_32_ (Table 1), undergoing modifications of the amino acids at positions 2, 3, 12, and 15, replaced by residues Lys, Gly, Leu, and Leu, giving rise to the peptide Ca-MAP1 (_NH2_-IKGIMSIVSKAFLGNLR-_COOH_). To demonstrate the sequence development, Table 1 shows the alignment of the paternal peptide candidalysin, and the peptide Ca-MAP1 furthermore shows the number of amino acid residues.

The percentage of structural homology of the Ca-MAP1 peptide with its respective candidalysin portion is 75% of identity, which represents the fully conserved amino acid residue. It has a 6.25% strong similarity, indicating the modification of amino acids with similar physical-chemical properties; 18.75% are differences demonstrating a complete change of amino acid residue.

The candidalysin sequence (_NH2_-SIIGIIMGILGNIPQVIQIIMSIVKAFKGNKR-_COOH_) has 32 amino acid residues, a net +4 charge, and hydrophobicity of 56.25%. The synthetic Ca-MAP1 (_NH2_-IKGIMSIVKAFLGNLR-_COOH_) has only 16 amino acid residues (50% of candidalysin length) and a reduced charge compared to the native +3 (Net charge of +3 is the most common net charge found in AMP deposited in APD [17]) and equal hydrophobicity of 56.25%. The physicochemical properties of these peptides were predicted using the Antimicrobial Peptide Database and are on display in Table 2 (https://aps.unmc.edu/prediction, accessed on 15 July 2022) and Heliquest (https://heliquest.ipmc.cnrs.fr/cgi-bin/ComputParams.py, accessed on 15 July 2022) [17,18].

### 2.2. Hemolytic Activity

After the synthesis of Ca-MAP1, the hemolytic activity of the Ca-MAP1 peptide was tested with murine-derived erythrocytes (Figure 2) and caused 91.09% hemolysis at a concentration of 72.7 µM., 71.95% at a concentration of 36.3 µM, 32.25% at a concentration of 18.1 µM, 5% at a concentration of 9.1 µM, and no hemolysis at concentrations below 4.5 µM.

### 2.3. Antibacterial Activity

Table 3 shows the minimum inhibitory concentration and minimum bactericidal concentration values of the Ca-MAP1 peptide against various resistant and sensitive Gram-negative and -positive bacteria. *E. coli*, *A. baumannii*, and *S. aureus* showed minimum inhibitory concentrations (MIC) of 9.1, 18.1, and 36.3 μM, respectively, and the minimum bactericidal concentration (MBC) resulted in the same concentration values. Ca-MAP1 did not demonstrate an inhibitory or bactericidal effect on the Gram-negative bacterium *P. aeruginosa* at the addressed concentrations, while ciprofloxacin presents activity with 6 μM to the same Gram-negative bacteria.

The antibacterial activity of the Ca-MAP1 presents a more effective activity with a lower concentration than the commercial antibacterial ciprofloxacin because all the ciprofloxacin concentrations of MIC/MBC were higher than the Ca-MAP1 concentrations in the cases of *A. baumannii* and *E. coli* ATCC and KPC.

To investigate the time of action of the Ca-MAP1 peptide, a membrane permeability assay with the dye Sytox Green was performed with *E. coli* KPC bacteria treated with the peptide at a concentration 30 times higher than the minimum inhibitory concentration [19,20]. The analyses show that in 15 min 84% of bacterial membrane permeabilization already occurs, and in 20 min Ca-MAP1 causes 98% permeabilization, as shown in Figure 3.

### 2.4. Anti-Neuroinflammatory Activity

To analyze the activity of Ca-MAP1 peptide in preventing LPS stimulation in BV-2 microglia cells, nitrite quantification was performed by Griess reaction methodology, as shown in Figure 4 [21].

The Ca-MAP1 peptide showed no significant cytotoxic effects for BV-2 cells in a concentration equal to and below 18.1 µM. The anti-inflammatory activity by inhibiting nitrite production in BV-2 cells that were stimulated by LPS was evaluated in non-cytotoxic concentrations for the BV-2 cell line. The concentrations that possessed the best anti-inflammatory activities in the proposed model were the concentrations of 9.1 and 18.1 µM, which inhibited 68.08 and 93.8% of nitrite production, respectively.

### 2.5. Anticancer Activity

Figure 5 shows the inhibition results for the cell viability of Ca-MAP1 peptide against RD, HeLa, and NCI-H292 tumor cell lines and MRC-5 fibroblast cell line after 24 h of treatment. Ca-MAP1 inhibited 46.33% of cell viability in the NCI-H292 cell at a concentration of 36.3 µM. There was also 30.04% inhibition of cell viability in the RD cell line, 9.89% inhibition of HeLa cellular viability and 21.27% inhibition of the MRC-5 cell line at the same concentration. The IC_50_ of the Ca-MAP1 peptide is 38.4 µM to the NCI-H292 cell line.

The commercial antineoplastic doxorubicin in the higher concentration (20 mg.mL^−1^ or 36.7 µM) showed inhibition of 27.61, 57.30, and 68.06% to the cellular viability of NCI-H292, RD, and HeLa, respectively. The normal cellular line MRC-5 presented an inhibition of 23.98% in the concentration of 36.79 µM. The values of IC_50_ of doxorubicin to the cellular line NCI-H292 is 28.1 mg.mL^−1^ or 51.8 µM with a selectivity index of 1.44.

### 2.6. Circular Dichroism

The Ca-MAP1 peptide was evaluated in the presence of water, 50% trifluoroethanol (TFE), and sodium dodecyl sulphate (SDS), which are hydrophilic, hydrophobic, and anionic environments, respectively for circular dichroism analyses. The band spectrum format (Figure 6) was characterized by the occurrence of two negative band values approximately between 208 and 222 nm and one positive band at 190 nm, which are characteristic of peptides with a flexible α-helix secondary structure in the presence of SDS and TFE [22,23]. In the presence of water, it is possible to observe a negative peak in the band at 199 nm, which is close to the characteristic peak at 190 nm for random coil [24,25].

### 2.7. Molecular Modeling

The three-dimensional structures of the Ca-MAP1 peptide were made by the I-TASSER server by comparison to analogous structures in the PDB database, where the server forms several possible models that each present a C-score, RMSD, and TM-score that indicates the reliability of the model based on the structural homology of existing sequences. The model with the best reliability was then modeled for visualization in the PyMol application (Figure 1). The predicted Ca-MAP1 three-dimensional structure validation scores obtain from I-TASSER [26], Prosa-web [27], and MolProbity [28,29] are displayed in Table 4.

### 2.8. Molecular Dynamics

The three-dimensional model of the candidalysin derivate peptide Ca-MAP1, generated by homology modeling on the I-TASSER server, was subjected to molecular dynamics simulations to analyze its behavior in an aqueous environment; the trajectory and topology of the Ca-MAP1 atoms indicate that there is a tendency to assume a random structure. The variation of the peptide backbone indicates an increase in RMSD, which evaluates the root mean square deviation of the position of the alpha carbons of the overlapping amino acid residues over simulation time, with a range of 0–0.9 nm (Figure 7A).

During the simulation, it is possible to observe moments in which there is an increase in the degree of decompression of the peptide. This may be due to the presence of hydrophilic residues that interact with water molecules, corroborating the increase in the diameter of the spin radius of the peptide as a whole, ranging from 0 to 0.95 nm; however, during much of the simulation, its diameter remained between approximately 0.65 and 0.8 nm (Figure 7B).

The root mean square fluctuation (RMSF) evaluates the side chain variation of each component amino acid residue of the peptide, demonstrating greater variation in the N-terminal and C-terminal regions, reaching a fluctuation of up to 0.6 nm for Ile^1^ and 0.55 nm for Arg^16^; terminal regions may vary more compared to amino acids within the sequence due to electrostatic interactions with the terminal amine and carboxyl, while more hydrophobic residues may remain more clustered and have less fluctuation (Figure 7C). Finally, the solvent-accessible surface area was calculated, indicating stability throughout the simulation, remaining approximately between 15 and 20 nm^2^ (Figure 7D).

## 3. Discussion

Studies have shown that the physicochemical properties of peptides are correlated so that modifications in the parameters can reflect significant changes to one or more of the others. Understanding and controlling these interrelationships may be the key to designing new peptides with greater potency and specificity [9,16].

The rational peptide design proposed comprehends the reduction of length of the candidalysin, which is linked to similar maintenance antimicrobial activity parameters, and decreases toxicity [30,31,32].

Analyzing the physical-chemical parameters of the deposited AMPs on APD demonstrates that the main frequency of charge is +3, and there is a hydrophobicity of 45 to 55% [17]. The exchange of Gln with Lys was to keep the charge in the highest frequency on the database [17]. The replacement of Iso with Gly was to facilitate the helix-helix packing and increase electrostatic and van der Waals interaction [33]. The substitution of Lys with Leu is connected to the increase of hydrophobic moment and helical structure stabilization [34,35].

The amino acid changes caused by the definition of amphipathicity provoke the rearrangement of the hydrophobic moment, increasing when compared to candidalysin. The increase of hydrophobic moment tends to be correlated to higher efficacy of cell binding and penetration [36,37,38].

The low percentage of hemolysis in antibacterial, antitumor, and anti-inflammatory peptides is a remarkable point in the development of new bioactive molecules due to the low amount or absence of undesirable side effects. The Ca-MAP1 peptide, when used at low concentrations (1 to 20 µM) presents no hemolytic activity. In contrast, in the literature, the cytolytic toxin candidalysin causes complete hemolysis of erythrocytes at a concentration of 10 µM [39,40].

The Ca-MAP1 peptide inhibited the growth of *K. pneumoniae* ATCC bacteria at concentrations of 18.1 μM and acted as a bactericide at the same concentrations. The *E. coli* KPC bacteria had growth inhibited at a concentration of 9.1 μM, but all bacterial cells were eliminated at concentrations of 36.3 μM. Ciprofloxacin inhibited growth at the concentration of 32 μg.mL^−1^ (96.5 μM) and was not shown to be bactericidal at the same concentration.

The antibiotic ciprofloxacin is not able to emit fluorescence caused by permeabilization of the *E. coli* bacteria membrane within twenty minutes, indicating that the time to cause its antibacterial activity is slow compared to the Ca-MAP1 peptide [41,42].

The Ca-MAP1 peptide was efficient in inhibiting Gram-negative bacterial growth at the lowest MIC of 9.1 µM, as in the case of *E. coli*, and at the same concentration there was relatively low hemolysis in blood cells. However, the release of lipopolysaccharide (LPS) by the lysis of Gram-negative bacteria causes sepsis and induces inflammation at picomolar concentrations [43,44].

Activated microglial cells play an important role in immune and inflammatory responses in the central nervous system and neurodegenerative diseases. Many pro-apoptotic pathways are mediated by signaling molecules that are produced during neuroinflammation. In glial cells, NF-κB, a transcription factor, initiates and regulates the expression of various inflammatory processes during inflammation, which is attributed to the pathology of various neurodegenerative diseases [45].

It is already known that LPS, which is an endotoxin in the outer membrane of Gram-negative bacteria, induces the systemic inflammatory response syndrome through toll-like receptor (TLR) signaling [4,46]. The binding of LPS to TLR4 on the surface of microglia activates several signal transduction pathways, including PI3K/AKT, MAPK, and mTOR, which ultimately lead to NF-κB activation. Activation of NF-κB then mediates the production of pro-inflammatory cytokines, chemokines, inducible nitric oxide synthase (iNOS), and COX-2, which together result in neuroinflammation [43,45]. Inhibition of NO production in LPS-induced BV-2 cells can be considered an effective treatment for CNS inflammation [47]. The NO inhibitory activities of the Ca-MAP1 peptide were evaluated in LPS-stimulated BV-2 cells, and the cytotoxic effects of these peptides were measured by MTT assay. Ca-MAP1 showed no cytotoxicity and inhibited the by-product NO (nitrite).

Chemotherapy is one of the most effective and widely used conventional treatments for cancer patients. Additionally, it is observed that 70% of cancer patients undergoing chemotherapy may develop cognitive problems during or after treatment, which affects their quality of life [48,49]. Chemotherapy is known to have adverse effects on brain function, causing dysfunctions in learning, memory, attention, motor activity, and executive function [50,51]. In addition, several studies show that tyrosine kinases, antimetabolites, microtubule inhibitors, and alkylating agents can induce neurotoxicity [52,53,54].

Doxorubicin belongs to the anthracycline class and is commonly used as a chemotherapy treatment for breast, hematologic, and lung cancer [55,56,57]. Doxorubicin exerts its antitumor effects through DNA insertion and topoisomerase II inhibition [58]. In this context, the anticancer activity of Ca-MAP1 was evaluated against tumor cell lines, RD, HeLa, and NCI-H292, and normal cell MRC-5 and BV-2 using doxorubicin as an antineoplastic standard. As we observed, the NCI-H292 strain, human lung cancer of the NCI family, the H292 type, was the one that showed the greatest capacity for resistance to doxorubicin [59,60] compared to the results of Ca-MAP1.

When comparing the IC_50_ values of doxorubicin and Ca-MAP1 in NCI-H292 cells, it showed that doxorubicin presented IC_50_ higher than that of Ca-MAP1. Moreover, Ca-MAP1 showed an IC_50_ 1.35 times less than DOX, demonstrating a higher cytotoxic activity forward NCI-H292. In addition, the derivate caused low cytotoxicity in comparison to doxorubicin to erythrocytes and human fibroblast lineage MRC-5 and neuronal microglia lineage BV-2 in ~18.1 µM.

The IC_50_ for MRC-5 was not calculated by the software because the Ca-MAP1 peptide did not demonstrate an effect on cellular viability close to fifty percent. In the case of the BV-2 cell line, the concentration of Ca-MAP1 for the NO assay was not cytotoxic. Therefore, it shows better activity than the positive control without *in vitro* deleterious consequences to the BV-2 cell line, MRC5 cell line, and erythrocytes.

After the activity assay, the next step was conformational analysis *in silico* and *in vitro*. The *in silico* studies on the creation of a structural homology-based model showed the possible formation of an α-helix, which can be validated via the Ramachandran plot [29]. The validation scores of the predicted three-dimensional structure demonstrate correct prediction based on structural homology [26,27]

Ca-MAP1 presented 92.86% of the amino acid residues in favored regions of the right-handed α-helix quadrant, 7.14% amino acid residue in the allowed region, and none in the disallowed region. In addition, the peptide Ca-MAP1 presents a Rama-Z of −1.85 ± 1.76, showing appropriated backbone geometry.

In the study of proteins by circular dichroism, it was possible to observe the characteristic spectra of peptide bonds at 190 nm, and in particular, in the bands around 208 and 222 nm, it is possible to estimate the formation of a helical structure, corroborated with the *in silico* studies [24,61].

In the circular dichroism, Ca-MAP1 demonstrated the occurrence of a positive band near 190 nm and two negative bands at approximately 208 and 222 nm. It is possible to assume that the peptide presents a secondary α-helical or 3_10_-helical structure that has close bands in the above values in the presence of TFE and SDS [24,25,62]. The formation of a helical structure is observed in the circular dichroism of the cytolytic peptide candidalysin in the HEPES buffer [15].

The broad negative band around 199 nm, and the fact that there is no other negative band in the analysis of circular dichroism in the presence of water, show that the peptide Ca-MAP1 can form a random coil [24,25], a result that is corroborated with the molecular dynamics of the peptide RMSD, RMSF, SASA, and Rg fluctuations in the presence of water [63]. More environments, such as SDS 30 mM and TFE 50%, need to be studied *in silico* with molecular dynamics to visualize if there is a corroboration between *in silico* and *in vitro* analysis.

Several studies have demonstrated the conformational changes in hydrophobic or hydrophilic environments for antimicrobial peptides through circular dichroism [64,65,66]. The CD analyses show that, in a hydrophobic environment, the conformational preference for Ca-MAP1 was α-helix, indicating that its structure favors the mechanism of action of the membrane permeabilization, as demonstrated in the Sytox Green assay. This result agrees with Migliolo et al., who demonstrated that an alanine-rich peptide presents helicoidal conformation and membrane disruption [67].

Other peptides in the literature have displayed similar activity and conformational behavior; antimicrobial activity, antibacterial in particular, is well described for cationic α-helix amphipathic peptides [68,69,70]. Mycotoxin peptide derivates have not been widely studied, but other peptide toxin derivates have demonstrated similar characterization results; animal venoms peptide is more widely used for the design of new drugs [69,70].

The Brazilian yellow scorpion (*Tityus serrulatus*) venom peptides TsAP-1 and TsAP2 possess different activities, the first being low hemolytic and bactericide at 120 to 160 µM. The second is more highly hemolytic and bactericidal at 5 to 10 µM. The increase of net charge by adding lysin to TsAP peptides increases hemolytic activity but dramatically increases the potency of antibacterial and anticancer effects, lowering the IC_50_ from 320 µM to 5 µM [64].

The exchange of amino acids to leucine and lysine has been effective in the creation of new peptide derivates in the venom peptide Hp1404 from the scorpion *Heterometrus petersii*; the derivates showed less hemolytic effect and antibacterial effect on multidrug-resistant *Pseudomonas aeruginosa* in the concentration of 0.78 to 25 µM, and all derivates had an amphipathic cationic α-helix in membrane-mimicking environments [69].

The cationic synthetic peptides Hp-MAP1 and Hp-MAP2, derived from the amphibian (*Hylarana picturata*) peptide toxin temporin-PTa, present antibacterial activity in concentrations ranging from 2.8 to 92 µM, without a hemolytic effect on erythrocytes, and in molecular dynamics present an α-helix in the presence of hydrophobic and anionic environments and can form interactions with saline and hydrogen bounds [70]. Molecular dynamics in the presence of membranes mimetic with 1,2-dipalmitoylsn-glycerol-3-phosphatidylglycerol (DPPG-anionic) and 1,2-dipalmitoyl-sn-lyco-3 phosphatidylethanolamine (DPPE-neutral) are needed to observe the interactions with the membrane phospholipids and the Ca-MAP1 peptide.

Other compounds from animal venom have equal biotechnological importance; the Brazilian snake *Bothrops moojeni* produces an important phospholipase, A2, which can be collected due to its anticancer effects, at a concentration of 9.2 µM, in many cancers cell lines, including lung mucoepidermoid carcinoma NCI-H292 [65]. Another example is the wasp and bee venom of the species *Vespa velutina*, which possess antibacterial activity in Gram-negative and -positive bacteria, and anti-inflammatory activity in LPS-induced BV-2 inflammation at concentrations of 0.5 to 20 µg.mL^−1^ [66].

## 4. Conclusions

In summary, the use of a rational design approach for the creation of Ca-MAP1, bioinspired on a fungal toxin, present here for the first time an initial characterization of viable antibacterial, anti-neuroinflammatory, and anticancer activity. The peptide Ca-MAP1 has the potential to be used for the control of exacerbated immune responses caused by bacterial infections that release toxins and by-products. It is also noteworthy that there is activity against a rare metastatic human lung cancer, NCI-H292, with a more discriminatory activity than the antineoplastic drug doxorubicin. More studies for the development of its characterization and comprehension of the underlying mechanisms of action are needed.

## 5. Materials and Methods

### 5.1. Rational Design

Candidalysin, a cytolytic mycotoxin peptide secreted from *C. albicans*, presents 32 amino acid residues, a net charge of +4, apolar residues of 56.5%, hydrophobicity of 0.679 in the Eisenberg scale, and hydrophobic moment of 0.408. Faced with the challenge of searching for information regarding candidates that are multifunctional with the potential to combat and control diseases, candidalysin presented activity on immunomodulatory and cytolytic human epithelial cells and is thus a peptide that is little characterized in the literature.

The strategy to design the Ca-MAP1 peptide (Appendix A) was guided by two steps and considered three requisites: (1) a shorter length comprising charge between +3 and +4; (2) apolar amino acid residue percentage between 40 and 60% with hydrophobicity in agreement with an Eisenberg scale above 0.400; (3) hydrophobic moment above 0.400 (Appendix A). The second step to construct Ca-MAP1 was to correlate the amino acid residue modification with amphipathicity organization, guided by helix diagrams and C-terminal structure stability with a helical conformation preference (Appendix A).

In agreement with the requisites described in the methodology, two sequences were found. One primary sequence with 14 and another with 16 amino acid residues, both located on the C-terminal functional hot spot region of the candidalysin (Appendix A). The peptide with a length of 14 was discarded due to a decrease in the net charge below the requirements.

Based on these two regions, amino acid glycine, leucine, and lysine frequencies, observed in the profiles of bacterial lantibiotics, plants cyclotides, and amphibian temporin antimicrobial peptide families, were used to create the derived peptide Ca-MAP1 [71]. The results demonstrated that the addition of amino acid preferences in positions 2, 3, 12, and 15, where the 2 and 3 positions are associated with amphipathicity and positions 12 and 15 are associated with helices stability, is in agreement with helices diagrams and theoretical models (Figure 1).

### 5.2. Peptide Synthesis

The peptides were synthesized by the solid-phase method using 9-fluorenyl-methoxycarbonyl chemistry [72], purified by reverse-phase high-performance liquid chromatography (RP-HPLC) to >98% purity on an acetonitrile/H_2_O-TFA gradient, and confirmed by electrospray ionization mass spectrometry by Aminotech Company (Sorocaba, Brazil). The Ca-MAP1 peptide was solubilized in Milli-Q ultrapure water to create a stock solution, which was stored in a −20 °C freezer and used for all assays.

### 5.3. Hemolytic Assay

The hemolytic assay was developed according to Kim et al. (2005) [73], with modifications. Red cells were washed three times with 50 mM phosphate buffer (PBS); pH 7.4. peptide solutions were added to the erythrocyte suspension (1% by volume) at a final concentration ranging from 4 to 128 μg.mL^−1^, which can be expressed at molar concentrations of 2.2 to 72.7 μM, in a final volume of 100 μL. The samples were incubated at room temperature for 60 min. Hemoglobin release was monitored by measuring the absorbance of the supernatant at 415 nm. Zero hemolysis (blank) was determined with red cells suspended in the presence of 50 mM PBS, pH 7.4, while a 1% (by volume) aqueous solution of Triton X-100 was used as a positive control (100% red cell lysis). This experiment was approved by CEUA under number 014/2018.

### 5.4. Minimum Inhibitory Concentration (MIC) and Minimum Bactericidal Concentration (MBC) Assays

The MIC assays were performed against strains of *Acinetobacter baumannii*, *Pseudomonas aeruginosa*, *Escherichia coli* (ATCC 25922 and KPC 001812446), *Klebsiella pneumoniae* (ATCC 13883), and *Staphylococcus aureus*. The bacteria were plated on Mueller Hinton agar (MHA) plates and incubated at 37 °C overnight. After this period, three isolated colonies of each bacterium were inoculated into 5 mL of Mueller Hinton broth (MHB) and incubated at 200 rpm at 37 °C overnight. Bacterial growth was monitored by a spectrophotometer at 600 nm.

MIC tests were performed by the 96-well microplate dilution method at a final bacterial concentration of 2–5 × 10^5^ CFU.mL^−1^. The peptides were tested at concentrations ranging from 4 to 128 μg.mL^−1^, which form the molar concentrations of 2.2 to 72.7 μM for Ca-MAP1. Ciprofloxacin was used as a positive control, due to its capability to be active against all of the clinically isolated bacteria used in this study; other tested antibiotics are displayed in Appendix A. The positive control was used in the same concentrations as the peptides, while the bacterial suspension in MHB was used as a negative control. The microplates were incubated at 37 °C for 18 h, and readings were taken on a Multiskan Go microplate reader (Thermo Scientific: Waltham, MA, USA) at 600 nm after the incubation time. MIC was determined as the lowest peptide concentration at which there was no significant bacterial growth.

Evaluation of the minimum bactericidal concentration (MBC) was dependent on the MIC results. Three replicates of 10 μL were taken from the microplate wells, plated in MHA, and incubated at 37 °C for 24 h. The MBC was determined as the lowest peptide concentration at which no bacterial growth was detected. All experiments were performed in triplicate.

### 5.5. E. coli Membrane Permeabilization

Membrane permeability was investigated as described by Mohanram and Bhattacharjya (2016) [20] and modifications proposed by Almeida et al. (2021) [19]. In the assay, a suspension of *E. coli* KPC + 001812446, grown in MH broth for 18 h at 37 °C, was prepared with OD_600 nm_ at 0.5 in 10 mM sodium phosphate buffer, pH 7.0. Then, 280 µL of the bacterial suspension was transferred to 96-well black microplates, where 10 µL of Sytox Green at 30 µM was added and incubated for 10 min at 37 °C. Subsequently, 10 µL of the peptide Ca-MAP1, at a concentration 30 times the MIC, was added to each well, and the kinetic assay was performed for 50 min, with readings every 5 min. The assay was performed with fluorescence readout, excitation at 485 nm, and emission at 520 nm in a Varioskan Lux microplate reader (Thermo Scientific: Waltham, MA, USA). The negative control of membrane damage was performed with *E. coli* KPC + 001812446, incubated with 10 µL of 10 mM sodium phosphate buffer, pH 7.0. Three independent experiments were performed, in triplicate.

### 5.6. Cell Cultures

In this study, we used diploid human fetal lung cell lineage (MRC-5), murine microglia cells (BV-2), human rhabdomyosarcoma cells (RD), cervical cancer cells (HeLa), and mucoepidermoid lung carcinoma (NCI-H292), of which MRC-5, RD, HeLa, and NCI-H292 cells were acquired from the Cell Culture Center of the Adolf Lutz Institute (São Paulo-SP), and BV-2 cells were acquired from the Rio de Janeiro Cell Bank. All cell lines were stored in liquid nitrogen cryopreservation at a temperature of approximately −196 °C at the Universidade Católica Dom Bosco (UCDB: Campo Grande, MS, Brazil). MRC-5, BV-2, RD, NCI-H292, and HeLa cells were cultured in the Immunology Laboratory (UCDB) in DMEM high glucose medium (for MRC-5 and RD) and RPMI-1640 (for BV-2, NCI-H292, and HeLa), respectively, supplemented with 10% fetal bovine serum (FBS), 100 U.mL^−1^ penicillin, and 100 µg.mL^−1^ streptomycin (Gibco: Waltham, MA, USA) at 37 °C in an incubator at 5% CO_2_.

### 5.7. Cell Viability Test Using MTT Methodology

To verify whether the peptide derivate inhibits anticancer activity, the viability of NCI-H292, HeLa, and RD cancer cells and normal cells MRC-5 and BV-2 were evaluated according to a method adapted from Mosmann (1983) [74], based on the enzymatic reduction of 3-(4,5-demethylthiazol-2-yl)-2,5-diphenyltetrazolium bromide (MTT: Sigma-Aldrich; St Louis, MO, USA) to formazan crystals. NCI-H292, HeLa, RD, MRC-5, and BV-2 cells were plated at 1 × 10^4^ cells. well^−1^ in 96-well microplates and treated with 100 µL of different molar concentrations of Ca-MAP1 (0.5, 1.1, 2.2, 4.5, 9.1, 18.1, 36.3 μM) for 24 h. Culture medium was used as a negative control. After the incubation period, the supernatant was removed and 100 μL of MTT solution (1 mg.mL^−1^ diluted in culture medium) was added to the cells. After 4 h of incubation, the formazan crystals were resuspended with 100 μL of dimethyl sulfoxide (DMSO) and read at 570 nm on a Thermo scientific reader (MultiSkan Go Model) [74]. The commercial antineoplastic doxorubicin was used as a positive control, with concentrations in serial dilution ranging from 0.3 to 20 mg.mL^−1^, which can be converted to micromolar concentrations ranging from 0.5 to 36.7 µM. Three independent experiments were performed in triplicate. Cell viability was calculated from the following formula:$$\mathrm{Cell}\;\mathrm{viability}\;\left(\%\right)=\left(\mathrm{AbsSample}\;+\;\mathrm{AbsNegativeControl}\right)\times 100$$

### 5.8. Inhibition of Microglial Activation by LPS

BV-2 microglia strains were plated at a density of 5 × 10^5^ cells.mL^−1^ in 96-well plates, followed by adhesion for 24 h at 37 °C in a 5% CO_2_ atmosphere. After adhesion, the medium was removed and cells were stimulated with lipopolysaccharide-LPS (Final concentration of 1 µg.mL^−1^), together with Ca-MAP1 peptide treated at various concentrations (0.2, 0.5, 1.1, 2.2, 4.5, 9.1, 18.1 μM) in a final volume of 100 μL.well^−1^ of RPMI culture medium supplemented with 1% SFB. For the control experiment, cells were cultured with a culture medium and medium with LPS. The cells were then incubated for another 24 h at 37 °C, 5% CO_2_, and the cell supernatant was collected for NO analysis with adhered cells, and cell viability assay by the MTT method was performed. The production of nitric oxide was measured by the dosage of its most stable degradation product, nitrite, using the Griess reagent [21]. For the determination of NO production, 100 μL of cell supernatant was subjected to the reaction with an equal volume of Griess reagent. For the preparation of this reagent, solutions of naphthylethylenediamine chloride (0.1%) dissolved in water and 1% sulfanilamide dissolved in H_3_PO_4_ (5%) were used. Just before use, the solutions were added in a 1:1 ratio, forming the Griess reagent properly. After the 10-min incubation period, the samples were read in a microplate reader at 540 nm. The calculation of nitrite concentrations was performed based on standard curves using different concentrations of Nitrite (3.12 up to 200 μM). Three independent experiments were performed in triplicate.

### 5.9. Circular Dichroism (CD)

The analyses were performed on a Jasco J-1100 spectropolarimeter (Jasco Inc.: Hachioji-shi, TYO, Japan) using a quartz cuvette with a 1 mm optical path. The spectrum from 260 to 185 nm was collected with steps at a resolution of 0.1 nm to 100 nm.s^−1^, at 25 °C, with an average of 5 accumulated scans for each spectrum. The Ca-MAP1 peptide was prepared in a stock solution of 120 µM and was incubated under different conditions at a concentration of 30 µM. The Ca-MAP1 peptide had its secondary structure analyzed in the presence of water, 50% trifluoroethanol (TFE) to mimic a hydrophobic environment, or 30 mM SDS, an amphipathic compound that forms micelles mimicking an anionic environment. The data were converted to molar ellipticity (θ), according to the equation:$$\left[\theta \right]=\frac{\theta }{10\;*\;C\;*\;l\;*\;\mathrm{nr}}$$
where θ is the ellipticity measured in milliseconds, C is the peptide concentration (M), l is the path length of the cuvette, and n_r_ is the number of amino acid residues. The fractional alpha-helix content, *fH*, was estimated using the equation:$$fH=\frac{\theta 222-\;\theta C}{\theta H\;-\;\theta C}$$
where θC = 2220 − 53, θH = (250 T − 44,000) (1 − 3/n), where T is the temperature in celsius and n is the number of amino acid residues in the peptide. The θC and θH values represent, respectively, the threshold values of average ellipticity at 222 nm (θ222) for a disordered and alpha-helix conformation.

### 5.10. Comparative Modeling and Validation

The theoretical three-dimensional models were built by fold recognition using template X-ray crystallography and the nuclear magnetic resonance of a similar primary sequence deposited in the PDB database through the I-TASSER server. Model validation was performed based on the statistical data of C-score, Z-score, and root mean square deviation or root mean square deviation of alpha-carbon position generated by the I-TASSER server. ProSA-web was used to calculate the overall quality score for the models based on score results within the ranges for native proteins [26,27]. The visualization of the predicted model was visualized by the PyMOL version 2.0 program [26,75]. The possible dihedral angles psi (Ψ) and phi (Φ) of each amino acid were calculated by MolProbity and observed in the Ramachandran plot [29,76].

### 5.11. Molecular Dynamics in Water

To perform the molecular dynamics simulations, the .pdb file model was generated from the modeling analyses of the I-TASSER server, and the calculations were performed by the program Gromacs version 5.0.4 (Groningen, The Netherlands) [77,78]. Molecular dynamics consists of three main stages; initially, assembly and parameterization of the system are performed, where a cubic box is generated in a vacuum, and it is filled with the solvent, composed of water molecules in the simple point charge (SPC) model and ions of Na^+^ and Cl^−^ to neutralize the total charge of the system. After the solvation is performed, the energy minimization of the system promotes the lowest free energy of the molecules; after this process, heating at a temperature of 310 Kelvin and pressurization of the system is carried out. After the parameterization of the system, the simulation calculations of the atomic movement and topology of the atoms of the system are performed based on molecular interactions according to the Newtonian mechanics of classical physics. Finally, after performing the calculations, the data analysis is carried out, which generates the values of RMSD, RMSF, spin radius, and solvent access area. All experiments were performed in triplicate.

### 5.12. Statistical Analysis

The statistical significance of the experimental results was determined by one-way Student’s *t*-test or one-way analysis of variance (ANOVA), followed by the Dunnett test. Values of *p* < 0.05 were considered statistically significant. GraphPad Prism version 8.0 was used for all statistical analyses.

## Figures and Tables

**Figure 1 toxins-14-00696-f001:**
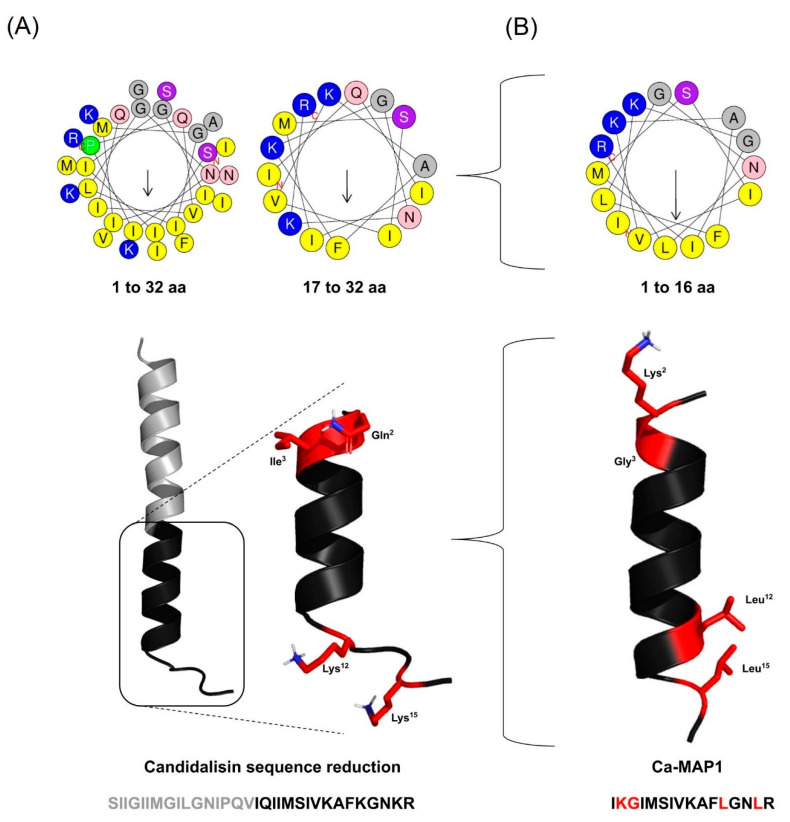
The strategy of rational design of the Ca-MAP1 peptide derivate from candidalysin: (**A**) Three-dimensional structure of candidalysin and reduced sequence portion of candidalysin with AMP physical-chemical characteristics with primary sequence and helical wheel projections; (**B**) Ca-MAP1 three-dimensional structure with primary sequence and helical wheel projection. The black amino acid residues in the three-dimensional structure are conserved, and the red is the changed amino acids. Legend: Positively charged amino acid residues are blue, negatively charged are red, hydrophobic aliphatic or aromatic are yellow/gray, and uncharged polar ones are pink/purple. The arrows within the diagram represent the hydrophobic moment.

**Figure 2 toxins-14-00696-f002:**
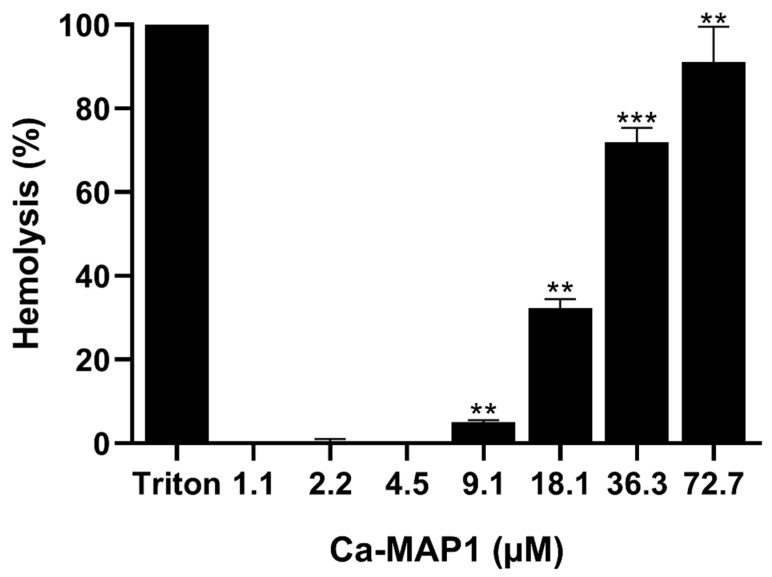
The hemolytic percentage from murine erythrocytes challenged with the Ca-MAP1 peptide in different micromolar concentrations. All experiments were performed in triplicate. Values of *p* ≤ 0.01 and *p* ≤ 0.0001 represent ** and ***, respectively. The *p* values were compared with cells treated with control.

**Figure 3 toxins-14-00696-f003:**
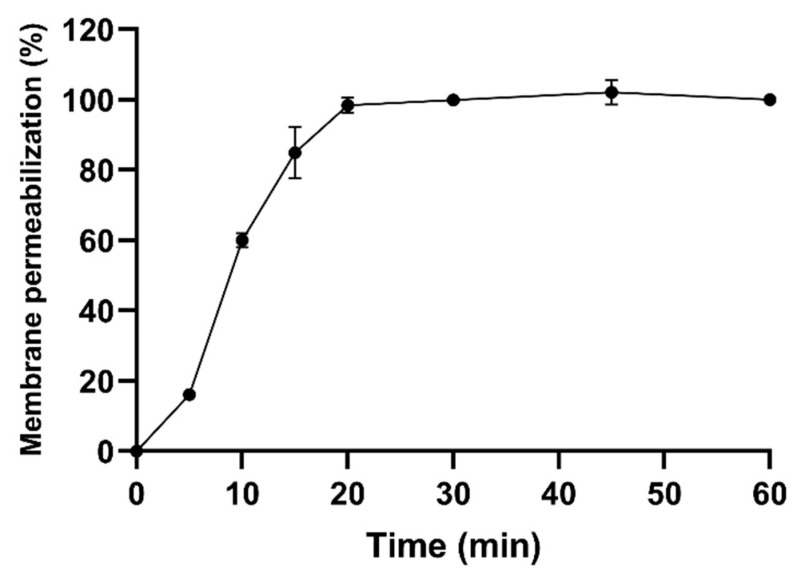
Effect on the permeabilization of E. coli bacterial membrane by Ca-MAP1 peptide over time, assed with Sytox Green dye assay. Three independent experiments were performed, in triplicate.

**Figure 4 toxins-14-00696-f004:**
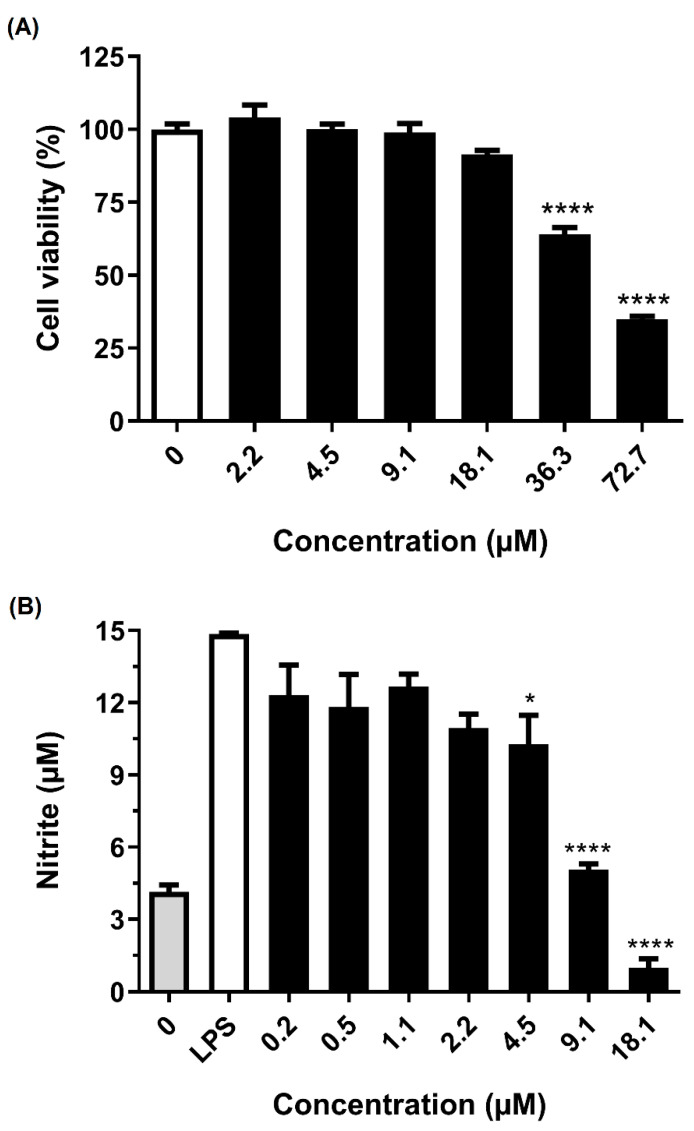
Effect of Ca-MAP1 peptide on (**A**) cellular viability of murine microglia BV-2 cell line and (**B**) NO production on BV-2 cell line stimulated with *E. coli* LPS. Three independent experiments were performed, in triplicate. Values are mean ± P.D.M. of three repetitions. Values of *p* ≤ 0.05 and *p* ≤ 0.0001 represent * and ****, respectively. The *p* values were compared with cells treated with control.

**Figure 5 toxins-14-00696-f005:**
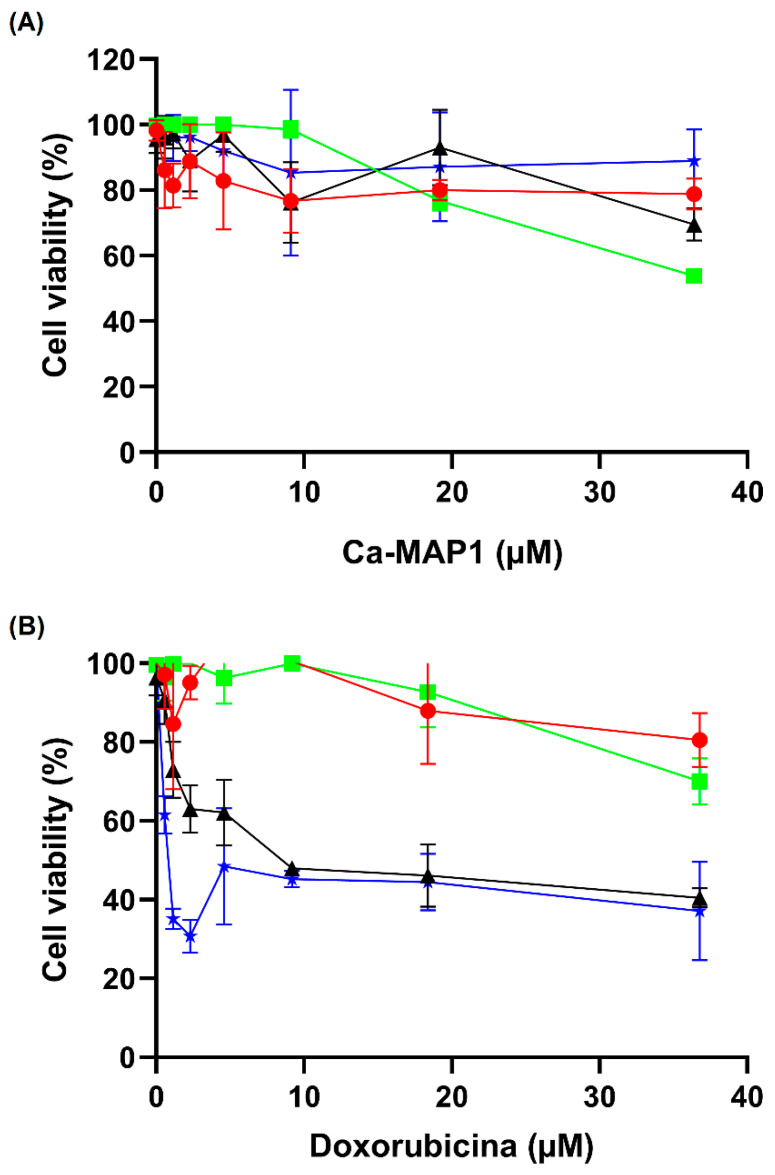
The cytotoxic effect of (**A**) Ca-MAP1 and (**B**) doxorubicin on the cell viability of cancer cell lines RD, HeLa, and NCI-H292, and the normal healthy MRC-5 cell line was evaluated by the MTT method. All experiments were performed in triplicate. Values are means ± P.D.M. of three repetitions. *p* > 0.0001 compared with cells treated with the control vehicle. Legend: Red circle = MRC-5, Green square = NCI-H292, Black triangle = RD, Blue star = HeLa.

**Figure 6 toxins-14-00696-f006:**
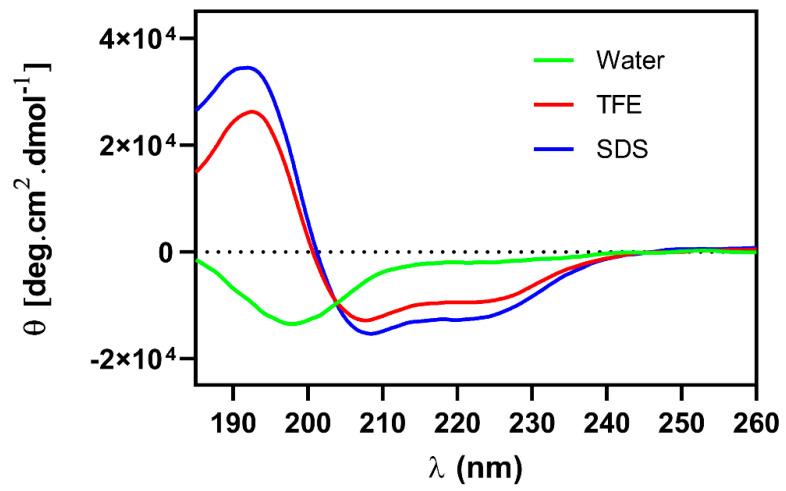
Circular dichroism analysis spectrum of the secondary structure of Ca-MAP1 peptide in the presence of water, TFE (50%), and SDS (30 mM), with a spectrum ranging from 185 nm to 260 nm.

**Figure 7 toxins-14-00696-f007:**
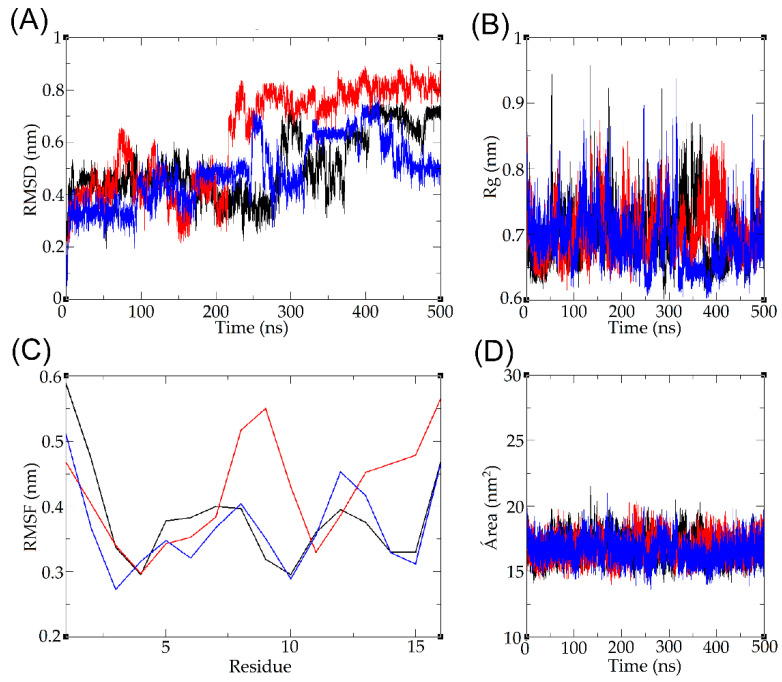
Molecular dynamics simulation result of Ca-MAP1 peptide in water, highlighting (**A**) root mean square deviation, (**B**) spin radius, (**C**) root mean square fluctuation, and (**D**) solvent accessible surface. The colors represent the triplicate of each parameter.

**Table 1 toxins-14-00696-t001:** Alignment between primary sequences of candidalysin and Ca-MAP1, showing in bold the portion used from the parental cytolytic toxin candidalysin as a framework for the design of Ca-MAP1, and the number of amino acid residues.

Name	Alignment	N° of Residues
Candidalysin	SIIGIIMGILGNIPQV**IQIIMSIVKAFKGNKR**	32
Ca-MAP1	----------------IKGIMSIVKAFLGNLR	16

**Table 2 toxins-14-00696-t002:** The *in silico* calculated physicochemical properties of the parental peptide candidalysin and its derivate peptide Ca-MAP1. Legend: <Z>, net charge; <H>, hydrophobicity; <µH>, hydrophobic moment; Da, Daltons.

Name	<Z>	Apolar (%)	<H>	<µH>	Mass (Da)
Candidalysin	+4	56.25	0.679	0.408	3464.05
Ca-MAP1	+3	56.25	0.607	0.631	1759.05

**Table 3 toxins-14-00696-t003:** Results of the minimal inhibitory concentration and minimal bactericidal concentration assays of Ca-MAP1 peptide and ciprofloxacin against Gram-negative and -positive bacteria. All experiments were performed in triplicate.

Bacteria	Ca-MAP1 (µM)		Ciprofloxacin (µM)	
	MIC	MBC	MIC	MBC
**Gram-negative**				
*Acinetobacter baumannii*	18.1	>72.7	96.5	>386.3
*Escherichia coli*	9.1	9.1	96.5	>386.3
*Escherichia coli* (KPC + 001812446)	9.1	9.1	386.3	96.5
*Klebsiella pneumoniae* (ATCC + 13883)	18.1	18.1	6	193.1
*Pseudomonas aeruginosa*	>72.7	>72.7	6	6
**Gram-positive**				
*Staphylococcus aureus*	36.3	36.3	6	6

**Table 4 toxins-14-00696-t004:** Summary of all model validation scores: C-score, RMSD, TM-score, Z-score, and Rama-Z from the predicted three-dimensional model on I-TASSER, ProSA-web, and MolProbity software version 4.5.1 (Duke University: Durham, NC, USA).

Name	C-Score	RMSD	TM-Score	Z-Score	Rama-Z
Ca-MAP1	0.05	0.6 ± 0.6Å	0.72 ± 0.11	−0.35	−1.85 ± 1.76

## Data Availability

Not applicable.

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
