# Peer review of "Evaluation of a Novel Synthetic Peptide Derived from Cytolytic Mycotoxin Candidalysin"

_toxins, 2022, doi:10.3390/toxins14100696_

Round 1

Reviewer 1 Report

This manuscript present study proposes to evaluate the anti-neuroinflammatory activity of synthetic peptides based on the protein sequence originating from the candidalysin of the fungus C. albicans. The language is well-organized and clear.  I suggest that this manuscript can be published after minor revisions.

1. Please explain why the cell viability are irregular as red and blue lines in fig. 5.

2. The format of the figures should be the same.

3. please add some discussions about the effects of Ca-MAP1 on normal cells.

Author Response

Reviewer #1:

This manuscript present study proposes to evaluate the anti-neuroinflammatory activity of synthetic peptides based on the protein sequence originating from the candidalysin of the fungus C. albicans. The language is well-organized and clear. I suggest that this manuscript can be published after minor revisions.

Response: The authors appreciate all the time dedicated to improving the work.

  1. Please explain why the cell viability are irregular as red and blue lines in fig. 5

Response: The graph in Figure 5 was organized in line so that we could analyze the cytotoxicity results more reliably, and also to make the graph more compact and reduced. The colors are only for better visualization. The irregularities refer to the standard errors between concentrations.

  1. The format of the figures should be the same.

Response: The authors agree, all the figures were standardized in TIFF format.

  1. please add some discussions about the effects of Ca-MAP1 on normal cells.

Response: The authors agreed and the discussion about the effects of Ca-MAP1 on normal cell lines such as BV-2, MRC5 and erythrocytes were more elaborated.

Now reads:

(Discussion: Page 12, line 324):

“Moreover, Ca-MAP1 showed an IC50 1.35x less than DOX, demonstrating a higher cytotoxic activity forward NCI-H292. In addition, the analog caused low cytotoxicity in comparison to doxorubicin to erythrocytes and in human fibroblast lineage MRC-5 and neuronal microglia lineage BV-2 in ~18.1 µM.”

The IC50 for MRC-5 wasn’t calculated by the software since the Ca-MAP1 peptide did not demonstrated effect on cellular viability close to fifty percent. In the case of the BV-2 cell line, the concentration of Ca-MAP1 for the NO assay was not cytotoxic. Therefore, shows better activity than positive control without in vitro deleterious consequences to the BV-2 cells line, MRC5 and erythrocytes.”

Reviewer 2 Report

The authors designed a peptide ‘Ca-MAP1’ based on the peptide sequence of candidalysin and tested its physicochemical properties. Ca-MAP1 showed haemolytic activity on murine erythrocytes, antibiotic activities, and E.coli membrane permeabilities at low concentrations, while anti-inflammatory activity and ‘anti-cancer’ activity were shown at higher concentrations. The structure prediction of ‘Ca-MAP1’ based on circular dichroism and molecular modelling suggests a helical structure. Several important issues to address.

1.       Title is somewhat misleading. The authors haven’t done the necessary experiments to suggest that Ca-MAP1 is actually anti-neuroinflammatory or anti-cancer. The readouts they use for this are basic/general and not specific.

2.       The question remains, why was Ca-MAP1 designed based specifically on candidalysin? Does Ca-MAP1 show toxicity profiles in comparison with candidalysin?

3.       It is not clear how Ca-MAP1 was designed. I understand that the physicochemical properties should be maintained but the authors do not explain why the length is set to 16 amino acids and how they've chosen the mutated positions.

4.       The authors chose four residues (Lys, Gly, Leu, and Leu) to generate Ca-MAP1. Have the authors considered mutating other residues than the four residues to reach at least 70 % similarity?

5.       Ca-MAP1 looks similar to candidalysin in the helix diagram prediction (Fig. 1). Have the authors compared candidalysin with Ca-MAP1 in their assays (as a control)? No comparison is currently in the ms.

6.       Most antibacterial peptides contain net positive charges. Regarding haemolytic activity/anti-bacterial activity/membrane permeabilization activity, the authors should test a scrambled version of Ca-MAP1 peptide to see whether the effect is due to the net charge alone.

7.       The authors should explain why they use TFE and SDS for their CD analysis. Also, the secondary structure prediction by CD showed two different types of structure in water and in SDS/TFE, respectively. What is the explanation of the CD curve of Ca-MAP1 in water?

8.       If Ca-MAP1 is not an a-helical in water (according to the CD), then Figure 6 might be misleading. Could the authors specify the caption regarding the buffer condition for molecular modelling?

9.       How was Ca-MAP1 peptide prepared for each assay? Was the peptide used in the study dissolved in water or in presence of SDS?

10.   The anti-inflammatory activity by inhibiting nitrite production in BV-2 cells that were stimulated by LPS was evaluated in non-cytotoxic concentrations for the BV-2 cell line. What about cytokine release? Is nitrite the ‘only’ inflammatory mediator from these cells? Also, molecules that are not affected by Ca-MAP1 need to be included to show specificity i.e. this could come from assessing cytokines.

11.   Please explain why Ciprofloxacin was chosen as control/reference peptide.

12.   Section 2.6 can be shorter. The information described can be condensed into a Table.

13.   Discussion section needs a bit of work, the main ideas are efficiently conveyed but the text is difficult to follow in places. 

14.   In the Discussion, the authors mention that low haemolytic activity is important for this family of peptides. I would appreciate if they could elaborate more on why low lysis is better.

15.   The Discussion could elaborate on how Ca-MAP1 compares with other peptides that show similar activity.

16.   Statistics are weak to non-existent. What does "significant" mean in the text?

17.   References should be checked. Notably, See Moyes et al. 2016, the original ms that described candidalysin. Many authors are excluded or omitted. Just look at PubMed for the correct author list. Please check all reference for such errors.

18.   Legends are way too minimal. They need lots more info.

Author Response

Reviewer #2:

The authors designed a peptide ‘Ca-MAP1’ based on the peptide sequence of candidalysin and tested its physicochemical properties. Ca-MAP1 showed haemolytic activity on murine erythrocytes, antibiotic activities, and E. coli membrane permeabilities at low concentrations, while anti-inflammatory activity and ‘anti-cancer’ activity were shown at higher concentrations. The structure prediction of ‘Ca-MAP1’ based on circular dichroism and molecular modelling suggests a helical structure. Several important issues to address.

Response: The authors appreciate all the time dedicated to improving the work.

  1. Title is somewhat misleading. The authors haven’t done the necessary experiments to suggest that Ca-MAP1 is actually anti-neuroinflammatory or anti-cancer. The readouts they use for this are basic/general and not specific.

Response: The authors agree and the title was remodulated to best fit the performed characterization of the peptide.

Now reads:

(Title: Page 1, line 2):

 “Evaluation of novel synthetic peptide derived from cytolytic mycotoxin candidalysin”

  1. The question remains, why was Ca-MAP1 designed based specifically on candidalysin? Does Ca-MAP1 show toxicity profiles in comparison with candidalysin?

Response: The authors agree, to make this issue clearer modifications on the material and methods were made. The reviewer’s observation is consistent however Ca-MAP1 is a peptide derived from candidalysin presenting its own characterization for being a new molecule. The confusion in interpretation is probably due to the use of the word "analog" which was replaced by "derived".

In comparison with the toxicity profile, Ca-MAP1 was evaluated on cancer cell lines and hemolytic effect presenting values of 36 and 72 µM, respectively. In contrast the candidalysin wasn’t tested for its cytotoxic effect, however, the total hemolysis activity found in the literature was of 10 µM.

Now reads:

(Material and Methods: Page 14, line 411):

“Candidalysin, a cytolytic mycotoxin peptide secreted from C. albicans, present 32 amino acid residues, net charge +4, amount of apolar residues of 56,25%, hydrophobicity of 0.679 in the Eisenberg scale and hydrophobic moment of 0.408. Faced with the challenge to search for information about candidates multifunctional with potential to combat and control diseases, candidalysin presented activity immunomodulatory and cytolytic human epithelial cells, thus being a peptide, little characterized in the literature.”

  1. It is not clear how Ca-MAP1 was designed. I understand that the physicochemical properties should be maintained but the authors do not explain why the length is set to 16 amino acids and how they've chosen the mutated positions.

Response: The author understands the reviewers’ issue, the rational design of Ca-MAP1 was constructed using various strategies such as hydrophobicity conservation, the net charge control in +3, increase hydrophobic moment and organizing the amphipathic and amphiphilic portion, those strategies were better elaborated in the material and methods section.

Now reads:

(Material and Methods: Page 14, line 417):

“The strategy to design the Ca-MAP1 peptide (Supplementary Material figure S1) were guided by two steps where the first to contemplate three requisites: (1) the shorter length comprising charge between +3 and +4; (2) apolar amino acid residues percentage between 40 to 60% with hydrophobicity in agreement Eisenberg scale above of 0.400 and the last requisite considered was (3) the hydrophobic moment above 0.400 (Supplementary Material table S1). The second step to construct Ca-MAP1 (Supplementary Material figure S1) was correlated the amino acid residues modification with the amphipathicity organization guided by helix diagram and C-terminal structure stability with helical conformation preference.

In agreement with the requisites described in the methodology two sequences were found. One primary sequence with 14 and other with 16 amino acid residues, both located on the C-terminal functional hot spot region of the candidalysin (Supplementary Material table S1). The peptide with length 14 was discarded due to de-crease the net charge below requirements.

Based in these two regions, the amino acids glycine, leucine and lysine frequency observed in the profiles of bacterial lantibiotics, plants cyclotides and amphibians temporins antimicrobial peptides families were used to create the derived peptide Ca-MAP1 [71]. The results demonstrated that the addition of amino acid preferences in the positions 2, 3, 12 and 15; where 2 and 3 positions are associated amphipathicity and the positions 12 and 15 are associated with helices stability in agreement helices diagram and theoretical models (Figure 1).”

  1. The authors chose four residues (Lys, Gly, Leu, and Leu) to generate Ca-MAP1. Have the authors considered mutating other residues than the four residues to reach at least 70 % similarity?

Response: The authors agree and the chosen of the four residues (Lys, Gly, Leu, and Leu) to generate Ca-MAP1 corroborate with the amino acid profile frequency for antimicrobial peptides deposited in the APD.

  1. Ca-MAP1 looks similar to candidalysin in the helix diagram prediction (Fig. 1). Have the authors compared candidalysin with Ca-MAP1 in their assays (as a control)? No comparison is currently in the ms.

Response: The authors agree. The candidalysin was not used in a control due to the derived peptide length modification, thus the goal of this work was introducing a novel peptide from candidalysin scaffold and compare with antibiotics communly used. To make this clearer a new figure was inserted in the place of Figure 1.

  1. Most antibacterial peptides contain net positive charges. Regarding haemolytic activity/anti-bacterial activity/membrane permeabilization activity, the authors should test a scrambled version of Ca-MAP1 peptide to see whether the effect is due to the net charge alone.

Response: The authors agree with the suggested idea and understand the importance of studying the net positive charge. However, the scramble version for new derived peptides will be a future manuscript.

  1. The authors should explain why they use TFE and SDS for their CD analysis. Also, the secondary structure prediction by CD showed two different types of structure in water and in SDS/TFE, respectively. What is the explanation of the CD curve of Ca-MAP1 in water?

Response: The authors understand the preoccupation and the use for different environment for CD analysis. To make the explanation more comprehendible in the manuscript the text was modified in the section CD analysis.

Now reads:

(Results: Page 8, line 198)

“The Ca-MAP1 peptide was evaluated in the presence of water, 50% trifluoroethanol (TFE) and sodium dodecyl sulphate (SDS) which are hydrophilic, hydrophobic and anionic environment, respectively for circular dichroism analyses.”

Now reads:

(Discussion: Page 13, line 359):

“Several relates demonstrated the conformational changes of in hydrophobic or hydrophilic environment for antimicrobial peptides through circular dichroism [68,69,70]. The CD analyses shows that in hydrophobic environment the conformational preference for Ca-MAP1 was α-helix indicating that its structure favor the mechanism of action of the membrane permeabilization as demonstrated in the Sytox Green assay. This result corroborates with Migliolo and collaborators which demonstrated that a peptide alanine rich present helicoidal conformation and membrane disruption [69].”

  1. If Ca-MAP1 is not an a-helical in water (according to the CD), then Figure 6 might be misleading. Could the authors specify the caption regarding the buffer condition for molecular modelling?

Response: The authors understand the preoccupation and the molecular modelling is performed by structural similarity of many PDB structures by treading methodology being the final model solved in the medium absence. The final model is used as input for molecular dynamic to compare the in silico and in vitro structural conformation reinforcing the results. The Figure 6 was removed and now is combined with the Figure 1 to help understand the strategy used for the rational design.

Now reads:

(Material and Methods: Page 17, line 574):

“The theoretical three-dimensional models were built by fold recognition using templates X-ray crystallography and nuclear magnetic resonance of similar primary sequence deposited in PDB database through the I-TASSER server.”

  1. How was Ca-MAP1 peptide prepared for each assay? Was the peptide used in the study dissolved in water or in presence of SDS?

Response: The question is interesting, Ca-MAP1 was solubilized in Milli-Q ultra-pure water to create a stock solution used for all the assays, the same protocol is used for the commercial antibiotic’s preparation.

Now reads:

(Material and Methods: Page 14, line 444):

“The Ca-MAP1 peptide was solubilized in Milli-Q ultrapure water to create a stock solution which was stored in a -20°C freezer and used for all assays.”

  1. The anti-inflammatory activity by inhibiting nitrite production in BV-2 cells that were stimulated by LPS was evaluated in non-cytotoxic concentrations for the BV-2 cell line. What about cytokine release? Is nitrite the ‘only’ inflammatory mediator from these cells? Also, molecules that are not affected by Ca-MAP1 need to be included to show specificity i.e. this could come from assessing cytokines.

Response: The authors agree with reviewer about the need to evaluate other inflammatory mediators such as TNF-α, IL-6 and IL-1, this would show us a possible anti-inflammatory mechanism of Ca-MAP1. However, the anti-inflammatory activity tested is an important preliminary result for Ca-MAP1. The evaluated concentrations that do not affect the cell viability of BV-2 cells shows the anti-inflammatory potential of Ca-MAP1. Because it is not interesting to have an anti-inflammatory that kills brain immune cells. Nitric oxide is an inflammatory mediator quantified by the formation of its metabolite’s nitrite (NO2-) and nitrate (NO3-) using the Griess reaction.

  1. Please explain why Ciprofloxacin was chosen as control/reference peptide.

Response: Others antibiotics were tested but they didn’t exhibit MIC/MBC, the results of the other antibiotic will be attached to the supplementary data. Ciprofloxacin was chosen due its ability to be active against all clinical isolated bacteria obtain and used in the MIC/MBC assay, thus, the ciprofloxacin was the only antibiotic capable of being compared with the Ca-MAP1 peptide in the same concentration.

Now reads:

(Material and Methods: Page 15, line 471):

“Ciprofloxacin was used as a positive control, because its capability to be active against all clinical isolated bacteria used in this study, other tested antibiotics are displayed in the Supplementary Material Table S3.”

  1. Section 2.6 can be shorter. The information described can be condensed into a Table.

Response: The authors agree and the information was condensed into the Table 4.

Now reads:

(Results: Page 9, line 217):

“The predicted Ca-MAP1 three-dimensional structure validation scores obtain from I-TASSER [26], Prosa-web [27] and MolProbity [28,29] were displayed in Table 4.”

(Table Caption: Page 9, line 221):

“Table 4. Summary of all model validation scores: C-score, RMSD, TM-score, Z-score, and Rama-Z from the predicted three-dimensional model on I-TASSER, ProSA-web and MolProbity software.”

Now reads:

(Results: Page 12, line 334):

“After the activity assay, the next step was the conformational analysis in silico and in vitro. The in silico studies on the creation of a structural homology-based model showed the possible formation of an α-helix that can be validated via the Ramachandran plot [63]. The validation scores of the predicted three-dimensional structure demonstrate correct prediction based on structural homology [26] [27].

Ca-MAP1, presented 92.86% of the amino acid residues in favored regions of the right-handed α-helix quadrant, 7.14% amino acid residue in allowed region and none are disallowed region. In addition, the peptide Ca-MAP1 present a Rama-Z of -1.85 ± 1.76, showing appropriated backbone geometry.”

  1. Discussion section needs a bit of work, the main ideas are efficiently conveyed but the text is difficult to follow in places.

Response: The authors agree, the discussion section was modified and divided in several paragraph along the manuscript to make the text more understandable.

  1. In the Discussion, the authors mention that low haemolytic activity is important for this family of peptides. I would appreciate if they could elaborate more on why low lysis is better.

Response: The author acknowledges the necessity to describe the importance of low hemolysis for novel drugs, to enlighten this issue, in the discussion segment that address hemolysis was more elaborated.

Now read:

(Discussion: Page 11, line 271):

“The low percentage of hemolysis in antibacterial, antitumor, and anti-inflammatory peptides is a remarkable point in the development of new bioactive molecules due to low or absence of undesirable side effect. Ca-MAP1 peptide when used at low concentrations (1 to 20 µM) no present hemolytic activity in contrast in the literature the cytolytic toxin candidalysin cause complete hemolysis of erythrocytes at a concentration of 10 µM [39,40].”

  1. The Discussion could elaborate on how Ca-MAP1 compares with other peptides that show similar activity.

Response: The authors agree and it was added information of other similar bioactive compounds and antimicrobial peptides in the discussion of the manuscript.

Now reads:

(Discussion: Page 13, line 366):

“Other peptides in literature have displayed similar activity and conformational behavior, the antimicrobial activity, antibacterial in particular is very described for cationic α-helix amphipathic peptides [65,66,67]. Mycotoxin peptide derivate aren’t widely studied, but other peptide toxin derivates have demonstrated similar characterization results, animal venoms peptide is more widely used for the design of new drugs [66,67].

The Brazilian yellow scorpion (Tityus serrulatus) venom peptide TsAP-1 and TsAP2 possess different activities the first being low hemolytic and bactericide at 120 to 160 µM, the second is more highly hemolytic and bactericidal at 5 to 10 µM. The in-crease of net charge by adding lysin in TsAP peptides increase hemolytic activity but dramatically increase the potence of antibacterial and anticancer effect, lowering the IC50 of 320 µM to 5 µM [68].

The exchange of amino acids to leucine and lysine has been effective to the creation of new peptide derivate inspired in the venom peptide Hp1404 from the scorpion Heterometrus petersii, the derivates showed less hemolytic effect, antibacterial effect on multidrug resistant Pseudomonas aeruginosa in concentration of 0.78 to 25 µM and all derivates has an amphipathic cationic α-helix in membrane mimicking environments [66].

The cationic synthetic peptide, Hp-MAP1 and Hp-MAP2 derived from the amphibian (Hylarana picturata) peptide toxin temporin-PTa, presents antibacterial activity in concentrations ranging from 2.8 to 92 µM without hemolytic effect on erythrocytes, and in molecular dynamics presents a α-helix in the presence of hydrophobic and anionic environment and can form interactions with saline and hydrogen bounds [67]. Molecular dynamic in the presence of membranes mimetic with 1,2-dipalmitoylsn-glycerol-3-phosphatidylglycerol (DPPG-anionic), 1,2-dipalmitoyl-sn-lyco-3 phosphatidylethanolamine (DPPE-neutral) are needed to observe the interactions with the membrane phospholipids and the Ca-MAP1 peptide.

Other compounds from animal venom are o equal biotechnological importance, the Brazilian snake Bothrops moojeni produce an important phospholipase A2 which can be prospected due its anticancer effects with concentration of 9.2 µM in many cancers cell lines, including the lung mucoepidermoid carcinoma NCI-H292 [69]. Other example is the wasp and bee venom of the species Vespa velutina that possess anti-bacterial activity in Gram-negative and -positive bacteria, and anti-inflammatory activity in LPS-induce BV-2 inflammation at concentrations of 0.5 to 20 µg.mL [70].”

  1. Statistics are weak to non-existent. What does "significant" mean in the text?

Response: The authors acknowledge the observation and necessity to better describe the statistical analysis used in this study. The “significant” meaning that the statistical analysis (Student’s t-test and one-way ANOVA) perform on GraphPad Prisma show P values lower than 0.05, displaying statistical difference when compared to the control.

Now reads:

(Material and Methods: Page 18, line 603):

“5.4. Statistical Analysis

The statistical significance of the experimental results was determined by one-way Student’s t-test or one-way analysis of variance (ANOVA) followed by Dunnett test. Values of P< 0.05 were considered statistically significant. GraphPad Prism version 8.0 was used for all statistical analyses.”

  1. References should be checked. Notably, See Moyes et al. 2016, the original ms that described candidalysin. Many authors are excluded or omitted. Just look at PubMed for the correct author list. Please check all reference for such errors.

Response: The authors agree. All references were carefully checked and corrected.

  1. Legends are way too minimal. They need lots more info.

Response: The authors agree and more information were added for all figures and tables.

Reviewer 3 Report

In this study, the authors developed a multiactivity peptide based on Candida toxin candidalysin and tested its activity against several bacteria and on tumor cell lines. The study is well designed and performed in a logical fashion. The manuscript is also well written, and the data is presented clearly. This study attains significance as the developed peptide can potentially be used to treat both bacterial infections and tumors, however, several studies need to be performed before the peptide-based drug can be approved. These initial studies will form a basis for further exploration of the efficacy of the ca-MAP1. There are a few minor corrections that need to be addressed.

1.       Please include the statistical analysis section in the Materials and Methods. Though statistics were performed for all experiments, the methodology used for analysis is not reported.

2.       Line 91: Change “demonstrate” to ‘demonstrating”

3.       Figure 2: Was parental peptide also tested here? If not including the parental peptide at 10 uM concentration will be an excellent positive control in future studies. As triton was used as a positive control in this study, the data is still valid for this figure for now.

4.       Lines 132-133: It is reported that P. aeruginosa did not show MIC and MBC. It is not clear if this bacterium is tested or not. If tested and not determined, does this mean the Ca-MAP1 is not effective against the bacteria? If so, this should be briefly discussed.

5.       Line 145: Remove ‘and’ from ‘bacteria and treated’

6.       Figure 5: panel b. Correct the spelling of ‘doxorubicin’

7.       Lines 231-234: What is the Rama-Z value of Ca-MAP1. Instead of mentioning that it is within the normal values, providing the exact value of Ca-MAP1 will be better.

8.       Lines 273-274: please reword the sentence.

Author Response

Reviewer #3:

In this study, the authors developed a multiactivity peptide based on Candida toxin candidalysin and tested its activity against several bacteria and on tumor cell lines. The study is well designed and performed in a logical fashion. The manuscript is also well written, and the data is presented clearly. This study attains significance as the developed peptide can potentially be used to treat both bacterial infections and tumors, however, several studies need to be performed before the peptide-based drug can be approved. These initial studies will form a basis for further exploration of the efficacy of the ca-MAP1. There are a few minor corrections that need to be addressed.

Response: The authors appreciate all the time dedicated to improving the work.

  1. Please include the statistical analysis section in the Materials and Methods. Though statistics were performed for all experiments, the methodology used for analysis is not reported.

Response: The authors agree and the suggestion was realized. The section of statistical analysis was added at the end of Materials and Methods as section 5.4.

Now reads:

(Material and Methods: Page 18, line 603):

“5.4. Statistical Analysis

The statistical significance of the experimental results was determined by one-way Student’s t-test or one-way analysis of variance (ANOVA) followed by Dunnett test. Values of P< 0.05 were considered statistically significant. GraphPad Prism version 8.0 was used for all statistical analyses.”

  1. Line 91: Change “demonstrate” to ‘demonstrating”

Response: The authors agree, the change of word was made to make the text more understandable.

  1. Figure 2: Was parental peptide also tested here? If not including the parental peptide at 10 µM concentration will be an excellent positive control in future studies. As triton was used as a positive control in this study, the data is still valid for this figure for now.

Response: The parental peptide was not used, the authors understand the use of the candidalysin as positive control in future studies, however, in this work we focus in to characterization of a novel synthetic peptide using the triton as positive control.

  1. Lines 132-133: It is reported that P. aeruginosa did not show MIC and MBC. It is not clear if this bacterium is tested or not. If tested and not determined, does this mean the Ca-MAP1 is not effective against the bacteria? If so, this should be briefly discussed.

Response: The authors agree with the reviewer and the term Nd was removed. Now, the Table 1 updated is represented by maximum concentration assayed for each bacterium represented by >72.7 µM.

  1. Line 145: Remove ‘and’ from ‘bacteria and treated’

Response: The authors agree. The change was made to make the text more comprehensive.

  1. Figure 5: panel b. Correct the spelling of ‘doxorubicin’

Response: The authors acknowledge and the modifications were made as suggested to correct the spelling of the antineoplastic drug doxorubicin.

  1. Lines 231-234: What is the Rama-Z value of Ca-MAP1. Instead of mentioning that it is within the normal values, providing the exact value of Ca-MAP1 will be better.

Response: The authors agree and the explanation about Rama-Z value for Ca-MAP1 was inserted in the text (Table 4) with all validation scores from the predicted three-dimensional structure.

  1. Lines 273-274: please reword the sentence.

Response: The authors agree and the alterations were made in order to make the text more coherent.

Now reads:

(Discussion: Page 11, line 257):

“The rational peptide design proposed comprehend the reducing of length of the candidalysin, which is linked to maintenance similar antimicrobial activity parameters, and decrease the toxicity [30,31,32].

Understand the AMPs physical chemical parameters of the deposited peptide on APD demonstrate that the mainly frequency of charge is +3 and of hydrophobicity 45 to 55% [17]. The exchange of Gln to Lys was to keep the charge in the highest frequency on the database [17]. Replacement of Iso to Gly was to facilitate the helix-helix packing and increase electrostatic and van der Waals interaction [33]. The substitution of Lys to Leu are connected to the increase of hydrophobic moment and helical structure stabilization [34,35].

As consequence of the amino acid changes caused by the definition of the amphipathicity it provokes the rearrange of the hydrophobic moment, increasing when compared to candidalysin. The increase of hydrophobic moment tends to be correlated to higher efficacy on cell biding and penetration [36,37,38].”

Round 2

Reviewer 2 Report

Authors have made a reasonable attempt to improve the ms, although extra experiments were not undertaken. This is acceptable.